# KPNB1 mediates PER/CRY nuclear translocation and circadian clock function

Yool Lee[1], A Reum Jang[2], Lauren J Francey[1], Amita Sehgal[2], John B Hogenesch[1]*

[1]Department of Systems Pharmacology and Translational Therapeutics, Institute for Translational Medicine and Therapeutics, Perelman School of Medicine at the University of Pennsylvania, Philadelphia, United States; [2]Department of Neuroscience, Howard Hughes Medical Institute, University of Pennsylvania Perelman School of Medicine, Philadelphia, United States

**Abstract** Regulated nuclear translocation of the PER/CRY repressor complex is critical for negative feedback regulation of the circadian clock of mammals. However, the precise molecular mechanism is not fully understood. Here, we report that KPNB1, an importin β component of the ncRNA repressor of nuclear factor of activated T cells (NRON) ribonucleoprotein complex, mediates nuclear translocation and repressor function of the PER/CRY complex. RNAi depletion of KPNB1 traps the PER/CRY complex in the cytoplasm by blocking nuclear entry of PER proteins in human cells. KPNB1 interacts mainly with PER proteins and directs PER/CRY nuclear transport in a circadian fashion. Interestingly, KPNB1 regulates the PER/CRY nuclear entry and repressor function, independently of importin α, its classical partner. Moreover, inducible inhibition of the conserved *Drosophila* importin β in lateral neurons abolishes behavioral rhythms in flies. Collectively, these data show that KPNB1 is required for timely nuclear import of PER/CRY in the negative feedback regulation of the circadian clock.

## Introduction

Nearly all organisms have circadian clocks, internal timing systems that anticipate and adapt physiology and behavior to daily changes in the light–dark cycle. In eukaryotes, the clock is an interlocked transcriptional–translational feedback loop, which drives rhythmic expression of both core clock genes that make up the oscillator and clock-output genes that drive biological rhythms. These include rhythms in the sleep wake cycle, body temperature, hormone secretion, and energy metabolism (*Reppert and Weaver, 2002*; *Bass and Takahashi, 2010*).

Many components and the molecular mechanism of the circadian clock are conserved between flies and mammals. Conserved components include the bHLH-PAS transactivators, CLOCK/NPAS2 (fly Clk) and BMAL1 (fly Cyc). This heterodimer binds E-box elements in the promoters of two families of transcriptional repressors, the *PER1/PER2/PER3* (fly *Per*) and *CRY1/CRY2* (fly *Cry*) genes. Once translated, PERs and CRYs (Per and Tim in flies) accumulate in a cytoplasmic complex and then translocate into the nucleus to repress the CLOCK-BMAL1 heterodimer, thereby repressing their own transcription (*Panda et al., 2002*). This primary negative-feedback loop is complemented by an ancillary loop, which while conserved, has different components in mammals and flies. In mammals, the ROR element binding repressors REV-ERBα/β and activators RORα/β/γ drive rhythmic expression of *BMAL1*. In flies, vrille (dVRI) or PAR-domain protein-1 (dPDP1) make up this ancillary loop and stabilize the *Drosophila* clock by driving rhythmic expression of *Clk* (*Yu and Hardin, 2006*). Further modulation of these feedback circuits occurs at the post-translational level as core clock proteins undergo phosphorylation, nuclear translocation, and protein degradation by several conserved kinases (CKIδ/CKIε, GSK3β; double-time [dDBT] and shaggy [dSGG]), phosphatases (PP2A; dPP2A), and E3 ubiquitin ligases (FBXL3, FBXL21; dSLMB1) (*Gallego and Virshup, 2007*; *Hirano et al., 2013*; *Yoo et al., 2013*).

*For correspondence:
hogenesc@mail.med.upenn.edu

**Competing interests:** The authors declare that no competing interests exist.

**Reviewing editor:** Achim Kramer

**eLife digest** Most organisms have an internal clock—known as the circadian clock—that regulates many aspects of their biology and behavior in roughly 24-hr long cycles. In animals, the core of the circadian clock is made of two 'activator' proteins and two 'repressor' proteins that inhibit the activators so that the levels of all four proteins in cells fluctuate over the cycle.

The activator proteins switch on the genes that encode the repressor proteins. This increases the production of the repressor proteins in an area of the cell called the cytoplasm. The repressor proteins then bind to each other and then move into the nucleus of the cell to inactivate the activator proteins. However, it was not clear how the repressor proteins move into the nucleus.

Lee et al. used a technique called 'RNA interference' to study the circadian clock in human cells and fruit flies. The experiments show that a protein called importin β enables the repressor proteins to move into the nucleus. Importin β directly interacted with only one of the repressor proteins (called PER). Previous studies have shown that importin β is able to interact with another protein called importin α, but Lee et al.'s results show that this interaction is not important for importin β's role in the movement of the repressor proteins.

Blocking importin β activity resulted in the loss of circadian rhythms in both human cells and fruit flies, which suggests that importin β performs the same role in many different animals. The circadian clock is disrupted in many cancers, so Lee et al.'s findings may also help to lead us to new treatments to fight these diseases.

KPNB1 is a nuclear import receptor that mediates docking of nuclear localization signal (NLS)-containing cargo bound to importin α to the nuclear envelope, thereby facilitating nuclear entry of the target protein (*van der Watt et al., 2013*). KPNB1 can also directly recognize the cargo and facilitate nuclear transport independently of importin α (*Lam et al., 1999*; *Forwood et al., 2001*; *Ghildyal et al., 2005*). In several intensive molecular analyses, PER1/2 and CRY1/2, have been found to contain a functional NLS for their nuclear entry and transcriptional feedback function (*Vielhaber et al., 2000*; *Miyazaki et al., 2001*; *Hirayama et al., 2003*; *Zhu et al., 2003*). Furthermore, nucleocytoplasmic shuttling of PER proteins, in conjunction with protein dimerization, phosphorylation and turnover, is critical for clock function (*Yagita et al., 2000*, *2002*). However, the specific carrier molecules responsible for nuclear translocation of the clock proteins have not been fully elucidated. Interestingly, a previous biochemical and cellular study reported that nuclear transport of mouse Cry2 is mediated by its direct interaction with importin α family members (RCH1, QIP-1, NPI-2) (*Sakakida et al., 2005*). However, the functional relevance of these factors in the clock was not evaluated (*Sakakida et al., 2005*). Moreover, the role of importin β was not investigated.

In recent studies, KPNB1 was identified as a key nuclear importin component of ncRNA repressor of nuclear factor of activated T cells (NFAT) (*NRON*) (lncRNA repressor of the NFAT) RNA-protein-scaffold complex containing several kinases, CKIε, GSK3β, and DYRK2 that regulate NFAT phosphorylation and nuclear translocation in human and *Drosophila* cells (*Gwack et al., 2006*; *Liu et al., 2011*; *Sharma et al., 2011*). Notably, most of these kinases are also well-known to regulate phosphorylation-related proteolysis and nuclear entry of circadian clock proteins. These data suggested that the clock may share a common molecular mechanism of nuclear translocation with NFAT.

Here, we tested this hypothesis and report that KPNB1 mediates circadian nuclear translocation and feedback repression activities of the PER/CRY complex through direct complex formation, independently of importin α. Loss of KPNB1 abolished not only rhythmic gene expression in human cells, but also circadian behavior in *Drosophila*. Collectively, these results highlight the evolutionarily conserved role of the KPNB1 in regulating nuclear translocation and function of the circadian clock.

## Results

### KPNB1 is a required for nuclear translocation of PER/CRY

The NRON complex is composed of a variety of proteins involved in signal transduction, proteolysis, and nuclear transport of NFAT (*Willingham et al., 2005*; *Sharma et al., 2011*). We noted that three of these components, CK1ε (*dco*, *Dbt*), GSK3β (*sgg*), and DYRK1a (*mnb*), are clock components in both

flies and mammals. Moreover, NFAT nuclear translocation has been suggested to be mediated by KPNB1 (*Willingham et al., 2005*; *Sharma et al., 2011*). Thus we hypothesized that KPNB1 would also be closely involved in nuclear entry of the PER/CRY complex and thus circadian clock function.

To test this, first, we explored how KPNB1 knockdown affects cellular localization of the core clock factors. Interestingly, in immunofluorescence (IF) analysis of cells expressing several flag-tagged clock proteins, knockdown of KPNB1 completely blocked (PER1, PER2), partly blocked (CRY1), or didn't significantly block (CRY2, REVERBα, CLOCK, CHRONO [*Anafi et al., 2014*]) nuclear accumulation (*Figure 1—figure supplement 1*). Similar to the flag-tagged proteins, microscopy and immunoblot analysis of cells expressing the Venus (V)-tagged clock proteins (PER1-V, PER2-V, CRY1-V, CRY2-V), KPNB1 knockdown markedly blocked PER1 and PER2 localization and slightly or negligibly effected CRY1 and CRY2 nuclear localization (*Figure 1A,B*). In many previous studies, PER and CRY proteins have been shown to work together (*Tamanini et al., 2005*). Hence, we looked into the effect of KPNB1 knockdown on the subcellular localization of various PERs/CRYs complexes using bimolecular fluorescence complementation (BiFC) (*Shyu et al., 2006*). The BiFC assay showed that KPNB1 depletion significantly impaired nuclear localization of various combinations of PER and CRY proteins (*Figure 1C*). Notably, nuclear accumulation of the PER2/CRY1 complex was totally blocked by KPNB1 knockdown (*Figure 1C*). In addition to ectopically expressed clock proteins, KPNB1 depletion substantially increased cytoplasmic accumulation of endogenous PER1/2 and CRY1/2 proteins thus reducing their relative levels in nuclei in most cells (*Figure 1D*, *Figure 1—figure supplement 2*). Taken together, these evidence show that KPNB1 serves as an integral nuclear transport receptor of the PER/CRY complex.

## KPNB1 regulates rhythmic nuclear localization and repressor activity of PER/CRY through direct complex formation

To investigate the detailed mechanism of the PER/CRY nuclear entry by KPNB1, we performed co-immunoprecipitation experiments with cells expressing the Venus-tagged PER and CRY proteins. KPNB1 interacted more strongly with PER1 and PER2 proteins than either CRY1 or CRY2 (*Figure 2*, *Figure 2—figure supplement 1A,B*). This suggested that PER proteins may play the leading role in PERs/CRYs nuclear translocation with KPNB1. In support of this, by IF KPNB1 co-localized with both cytoplasmic and nuclear PER2 (*Figure 2B*, *Figure 2—figure supplement 1C*). Correspondingly, KPNB1 strongly interacted with the endogenous PER2 in co-immunoprecipitation analysis using mouse liver tissue extracts (*Figure 2C*). To investigate the circadian role of KPNB1 in nuclear localization of PER/CRY, we prepared cytoplasmic and nuclear extracts from mouse liver tissues collected at 4 hr intervals for 24 hr in constant darkness. By immunoblot analysis, KPNB1 exhibited a circadian pattern of nucleocytoplasmic localization with its nuclear abundance peaking at circadian time (CT) 14–18, when the intensive nuclear accumulation of PER2 and CRY1 occurred (*Figure 2D*). Concomitantly, we observed the significant decrease of mRNA expression of E-box dependent clock genes (*Per1*, *Dbp*, *Nr1d1*, *Nr1d2*) targeted by the clock repressors in their nuclear accumulation phase (*Figure 2E*). Thus, these data clearly suggest that KPNB1 is responsible for circadian nuclear import and repressor activity of the PER/CRY complex in clock gene expression in mouse liver.

## KPNB1 regulates transcriptional repressor activity of PERs/CRYs and circadian gene expression independently of importin α (KPNA2)

To investigate the role of KPNB1 in clock function, we used RNAi to knockdown *KPNB1* in U2 OS cells and found substantially elevated *Per1* promoter activity, consistent with the hypothesis that this knockdown inhibited the localization and function of the PER/CRY complex (*Figure 3A*, *Figure 3—figure supplement 1*). In contrast, the parallel depletion of the other NRON complex nucleocytoplasmic transport proteins, *TNPO1*, *CSE1L*, did not significantly affect their repressor activities (*Figure 3A*, *Figure 3—figure supplement 1*, [*Willingham et al., 2005*]). In support of this, chromatin immunoprecipitation (ChIP) analysis targeting the first E-box region of human *PER1* (*hPer1*) promoter revealed significantly less recruitment of both PER1 and PER2 to the CLOCK/BMAL1 responsive element in the KPNB1-depleted U2 OS cells (*Figure 3B–D*). Moreover, the overall expression levels of several E-box dependent genes (*PER1*, *CRY1*, *DBP*, *REV-ERBβ*) were upregulated as a result of *KPNB1* knockdown (*Figure 3E*), consistent with loss of repressor function. Conversely, overexpression of *KPNB1* markedly down-regulated CLOCK/BMAL1-activated *PER1* transcription,

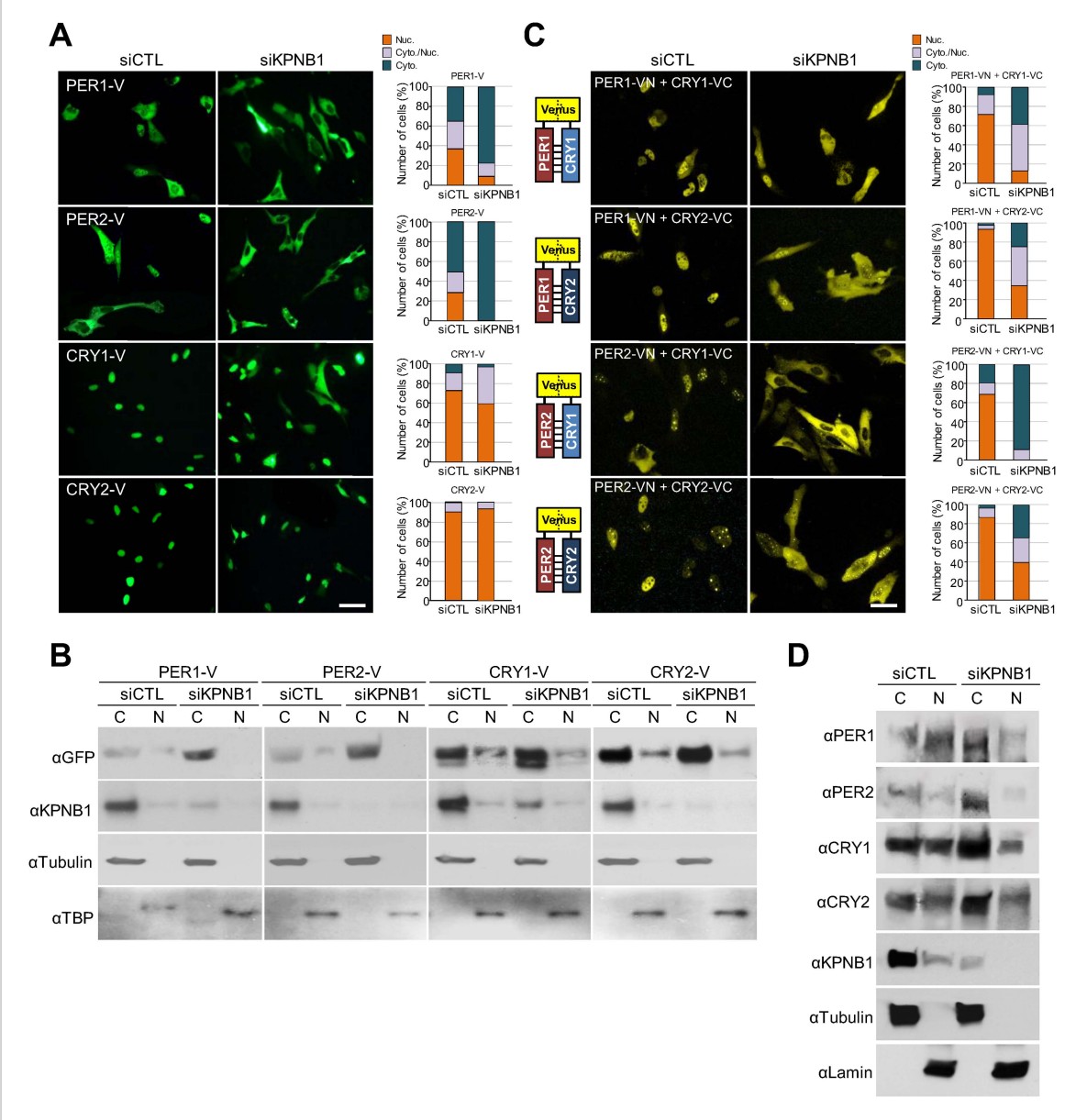

**Figure 1**. KPNB1 knockdown blocks nuclear translocation of PERs/CRYs complex. (**A**) Fluorescence microscopic analysis of the effect of KPNB1 depletion on subcellular localization of PER1, PER2, CRY1, and CRY2. Representative fluorescence images of U2 OS cells expressing Venus-tagged PER1, PER2, CRY1, and CRY2 (PER1-V, PER2-V, CRY1-V, CRY2-V) in the presence of control and *KPNB1* siRNA (left). Quantification of the subcellular distribution of the imaged cells (right). Subcellular localization is categorized as nuclear (Nuc.; orange), cytoplasmic and nuclear (Cyto./Nuc.; gray), and cytoplasmic (Cyto.; blue-green). More than 100 fluorescent cells for each of the images were evaluated. Scale bar: 30 μm. (**B**) Immunoblot analysis of nuclear and cytoplasmic extracts of U2 OS cells expressing PER1-V, PER2-V, CRY1-V, and CRY2-V in the presence of control and *KPNB1* siRNA using the indicated antibodies to GFP for Venus fused proteins (αGFP), KPNB1 (αKPNB1), Tubulin for cytoplasmic fraction (**C**) marker (αTubulin), and TATA box binding protein (TBP) for nuclear fraction (N) marker (αTBP). Representative images from two independent experiments are shown. (**C**) Bi-molecular fluorescence complementation analysis of the effect of KPNB1 depletion on subcellular localization of PER1/2-CRY1/2 dimeric complex in various combinations in U2 OS cells. Representative fluorescence images of cells expressing Venus N terminal (VN) or C-terminal (VC)-tagged PER1, PER2, CRY1, and CRY2 (PER1-VN, PER2-VN, CRY1-VC, CRY2-VC) as indicated combinations in the presence of control (siCTL) and *KPNB1* siRNA (siKPNB1) (left). Quantification of the subcellular distribution of the imaged cells (right). Subcellular localization is categorized as nuclear (Nuc.; orange), cytoplasmic and nuclear (Cyto./Nuc.; gray), and cytoplasmic (Cyto.; blue-green). More than fluorescent 100 cells for each of the images were evaluated. Scale bar: 30 μm. (**D**) Immunoblot analysis of nuclear and cytoplasmic extracts of U2 OS cells in the presence of control and *KPNB1* siRNA using the indicated antibodies for endogenous PER1 (αPER1), PER1 (αPER2), CRY1 (αCRY1), CRY2 (αCRY2), KPNB1 (αKPNB1), Tubulin for cytoplasmic fraction (**C**) marker (αTubulin), and Lamin for nuclear fraction (N) marker (αLamin). Representative images from three independent experiments are shown.

*Figure 1. continued on next page*

*Figure 1. Continued*

The following figure supplements are available for figure 1:

**Figure supplement 1**. Inhibition of KPNB1 alters nuclear localization of core clock repressor proteins.
**Figure supplement 2**. IF analyses of KPNB1 depletion effect on subcellular localization of endogenous PER1, PER2, CRY1, and CRY1.

probably by facilitating PER/CRY localization to the nucleus (*Figure 3—figure supplement 2*). Correspondingly, *KPNB1* overexpression significantly down-regulated endogenous mRNA expression of these E-box-regulated genes with no significant effect on *BMAL1* expression (*Figure 3E*).

Notably, the KPNB1 activity was not dependent on importin α, a well-known nuclear transport partner, as deletion of the importin α binding domain of KPNB1 did not significantly affect the

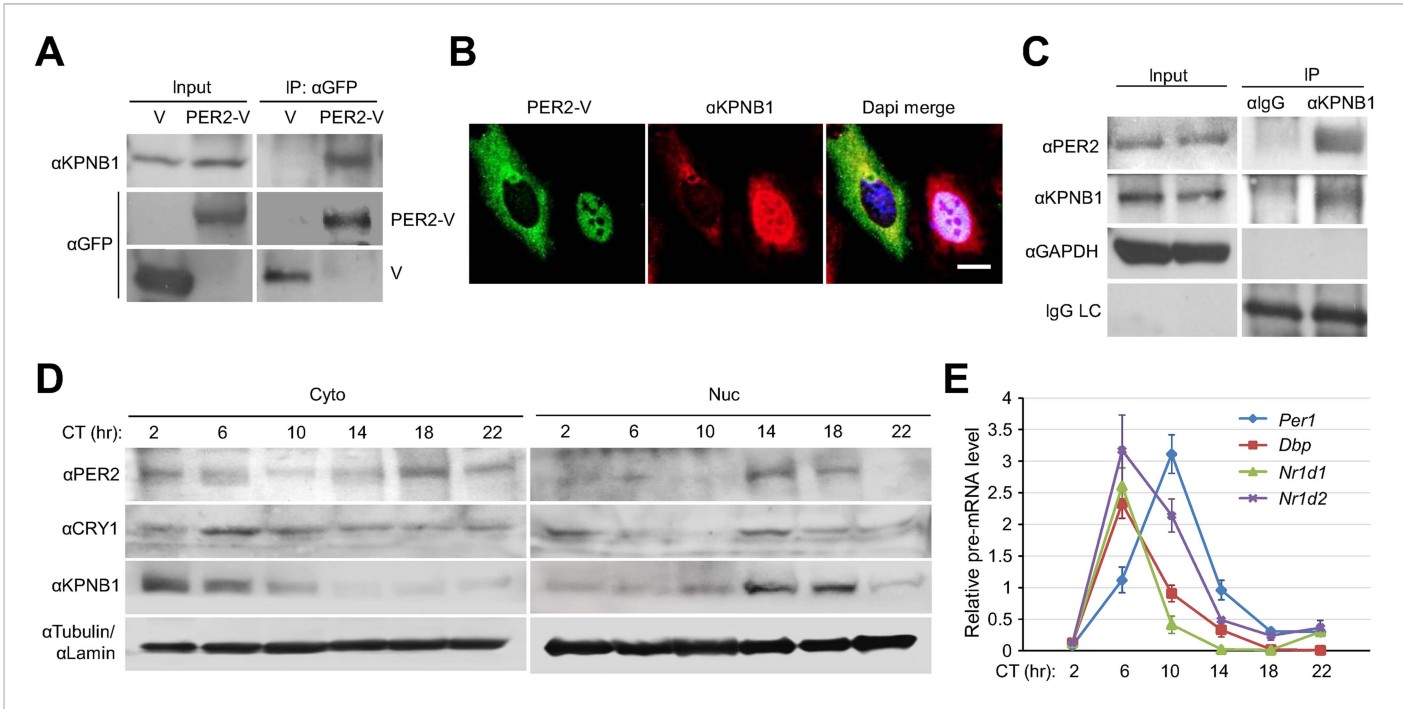

**Figure 2**. KPNB1 interacts with PER2 for circadian nuclear entry and repressor activity of PER/CRY. (**A**) Immunoprecipitation (IP) analysis of interactions of KPNB1 with PER2. U2 OS cells were transfected with Venus-tagged PER2, and then immunoprecipitated with anti-GFP antibody (IP: αGFP). The immunoprecipitates were analyzed by immunoblotting with anti-KPNB1 (IB: αKPNB1) or anti-GFP antibody (IB: αGFP). Representative images from three independent experiments are shown. (**B**) Immunofluorescence (IF) analyses of subcellular colocalization of PER2 with KPNB1. U2 OS cells expressing PER2-V were fixed and immunostained with antibody to endogenous KPNB1. The representative images were captured by fluorescence imaging microscopy using specific filter sets for FITC (green; PER-V), TRITC (Red; αKPNB1), and DAPI (Blue; Nuclei). See *Figure 2—figure supplement 1*. Scale bar: 10 μm. (**C**) Coimmunoprecipitation of PER2 with endogenous KPNB1. Liver extracts were immunoprecipitated (IP) with anti-KPNB1 or control IgG antibodies, and the immunoprecipitated proteins were probed with antibodies to PER2, KPNB1 as well as GAPDH as negative control and IgG light chain as a positive control for the IP. Representative images from three independent experiments are shown. (**D**) Immunoblotting analysis using cytoplasmic and nuclear extracts prepared from mouse liver tissues collected at 4 hr interval as indicated for 24 hr in constant darkness (CT: Circadian time). Anti-PER2, anti-CRY1, and anti-KPNB1 antibodies were used for detecting endogenous PER2, CRY1, and KPNB1 proteins. Anti-Tubulin antibody (αTubulin) for cytoplasmic fraction (Cyto) marker and anti-Lamin (αLamin) for nuclear fraction (Nuc) marker were used for loading controls respectively. Representative images from two independent experiments are shown. (**E**) Circadian expressions of endogenous clock gene mRNAs (*Per1*, *Dbp*, *Nr1d1*, *Nr1d2*) were determined by quantitative RT-PCR analysis of mouse liver tissue samples collected at 4 hr interval as indicated for 24 hr in constant darkness (CT; Circadian time). The data presented are the means ± S.E. of triplicate samples.
The following figure supplement is available for figure 2:

**Figure supplement 1**. KPNB1 interacts and colocalizes with PER2.

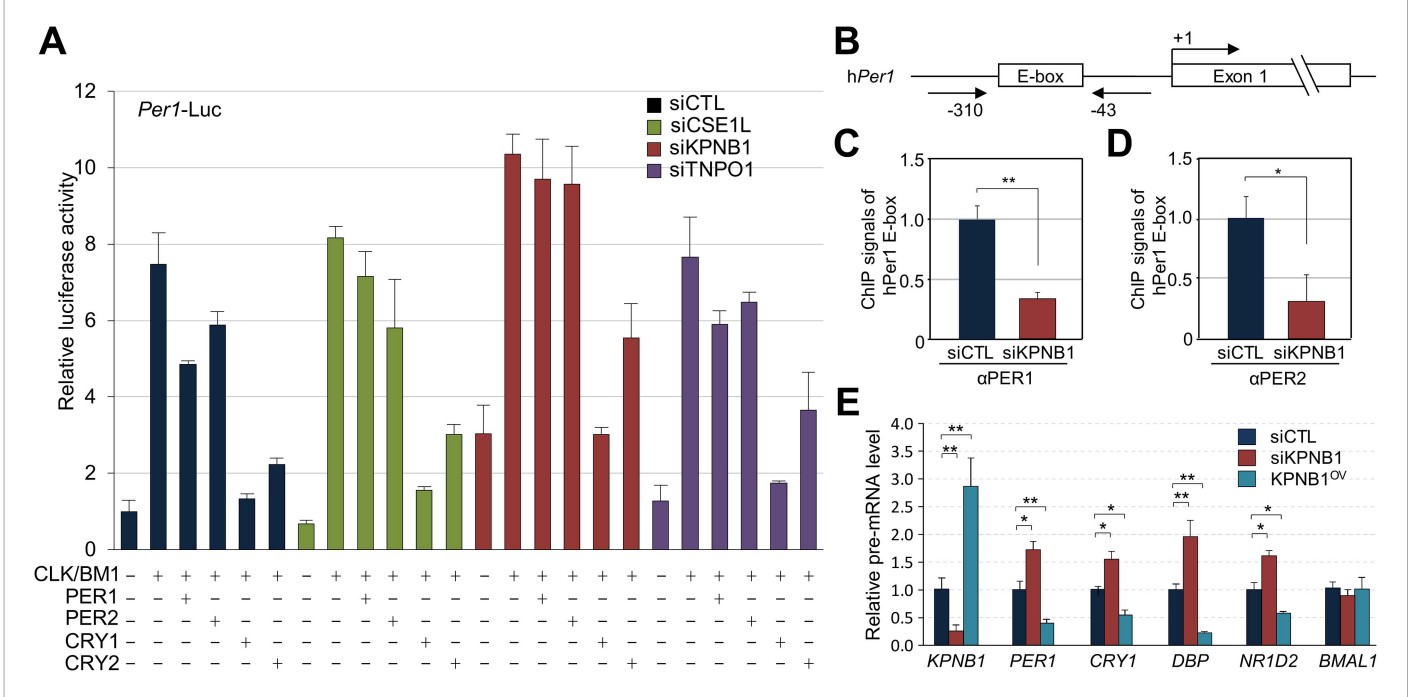

**Figure 3**. KPNB1 mediates PERs/CRYs-regulated transcription of E-box dependent clock genes. (**A**) HEK293T cells were transiently transfected with *Per1-Luc* reporter construct alone or cotransfected with plasmids expressing *CLOCK, BMAL1, PER1, PER2, CRY1, CRY2*, in the presence of control (siCTL: black), *CSE1L* (siCSE1L: light green), *KPNB1* (siKPNB1; gray red), and *TNPO1* siRNA (siTNPO1: violet) as indicated. After 24 hr, the cells were lysed and *Per1* promoter-driven luciferase activities were measured and normalized with pRL-TK activity. Results of one representative experiment of three independent experiments are shown. (**B**) Schematic diagram of the human *PER1* promoter and primers used for ChIP assay. (**C, D**) ChIP assay of PER1 or PER2 binding to the E box in *hPER1*. Control (siCTL: black) and KPNB1 siRNA (siKPNB1; gray red)-treated cells (U2 OS) were subjected to ChIP assays using anti-PER1 (αPER1) or anti-PER2 (αPER2) antibody. ChIP DNA samples were quantified by quantitative real-time RT-PCR. The data presented are the means ± S.E. of triplicate samples (**$p < 0.005$, *$p < 0.05$, by Student's *t*-test). (**E**) Quantitative real-time RT-PCR analysis of expression of endogenous *PER1, CRY1, DBP, REVERBβ*, and *BMAL1* mRNAs in control (siCTL: black), *KPNB1*-depleted (siKPNB1; gray red) or overexpressed (KPNB1$^{OV}$; blue green) cells (U2 OS). The data presented are the means ± S.E. of triplicate samples (**$p < 0.005$, *$p < 0.05$, by Student's *t*-test).

The following figure supplements are available for figure 3:

**Figure supplement 1**. Depletion of KPNB1 reduces repressional activity of PER and CRY proteins.

**Figure supplement 2**. Deletion of KPNA2, importin α binding domain is not required for regulatory function of KPNB1 in clock gene transcription.

transcriptional regulation (*Figure 3—figure supplement 2A–C*) (*Kutay et al., 1997*). Importantly, knockdown of *KPNB1* in U2 OS cells severely disturbed circadian bioluminescence reporter activity driven by the *Per2* promoter, whereas depletion of importin α1 (*KPNA2*) or importin α5 (*KPNA1*) did not affect the circadian rhythmicity (*Figure 4A,B*, *Figure 4—figure supplement 1A,B*, *Figure 4—figure supplement 3A,B*). These knockdown phenotypes correspond to previous reports showing that most of the other importin α isoforms were not critical for circadian rhythmicity (*Wu et al., 2009*; *Zhang et al., 2009*). In conjunction with these results, fluorescence microscopy analysis revealed that KPNA2 or KPNA1 depletion did not alter PER2 nuclear localization, which was completely blocked by KPNB1 depletion in U2 OS cells stably expressing PER2-Venus (*Figure 4—figure supplement 1C,D*, *Figure 4—figure supplement 3C,D*). On the other hand, KPNB1 depletion severely disrupted circadian transcription of several clock genes (*PER1, PER2*) in dexamethasone (Dex)-synchronized cells with altered nuclear accumulation of PER and CRY proteins (*Figure 4C,D*). Taken together, these data show that KPNB1 gates PER and CRY localization to the nucleus independently of importin α and plays an indispensable role in negative feedback regulation of the circadian clock in human cells.

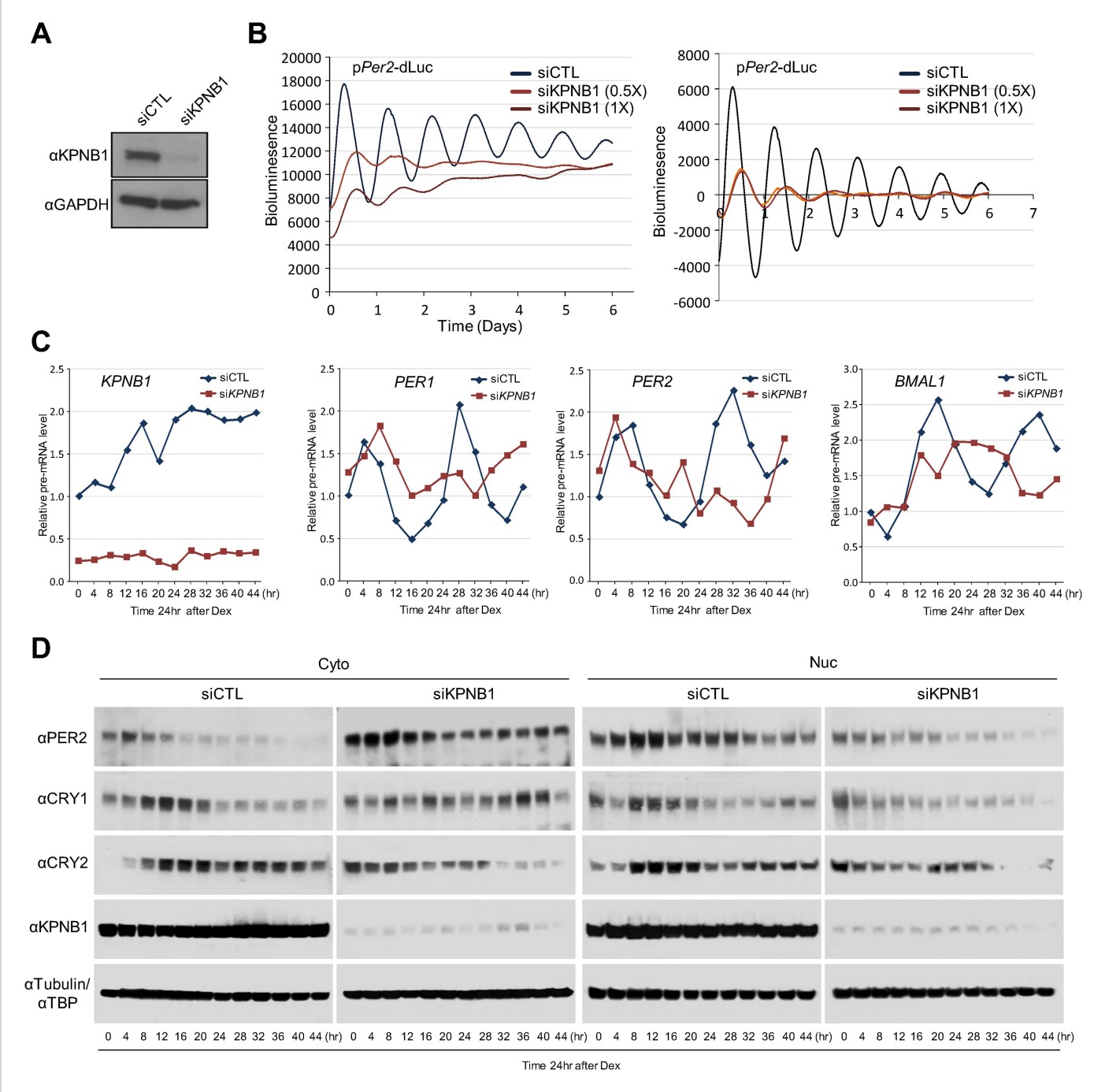

**Figure 4**. KPNB1 is required for circadian gene expression. (**A**) Immunoblot analysis of KPNB1 knockdown efficiency with anti-KPNB1 (αKPNB1) in control (siCTL) and *KPNB1* siRNA (siKPNB1)-treated U2 OS cells. Anti-GAPDH antibody (αGAPDH) was used as a loading control. (**B**) Bioluminescence recordings of dexamethasone (Dex)-synchronized control (siCTL) and increasing dose of KPNB1 siRNA (siKPNB1 0.5×/1×)-treated U2 OS cells expressing *Per2* promoter-driven destabilized luciferase (left; p*Per2*-dLuc). The Bioluminescence recordings were detrended by a 24-hr moving average subtraction (right). (**C**) Altered rhythmic expression of *KPNB1, PER1, PER2,* and *CRY1* transcripts by KPNB1 depletion. mRNA expressions of the target genes were determined by quantitative RT-PCR in control or KPNB1-depleted U2 OS cells over the course of 44 hr after 24 hr upon Dex treatment. The data are shown with the mean value of triplicate samples in each time point. (**D**) KPNB1 knockdown alters rhythmic nuclear accumulation of PER and CRY proteins. Immunoblotting analysis using cytoplasmic and nuclear extracts prepared from control or KPNB1-depleted U2 OS cells collected at 4 hr interval over the course of 44 hr after 24 hr of Dex treatment. Anti-PER2, anti-CRY1, anti CRY2, and anti-KPNB1 antibodies were used for detecting endogenous PER2, CRY1, CRY2 and KPNB1 proteins. Anti-Tubulin antibody (αTubulin) for cytoplasmic fraction (Cyto) marker and anti-TATA binding protein (αTBP) for nuclear fraction (Nuc) marker were used for loading controls respectively. Representative images from two independent experiments are shown.

*Figure 4. continued on next page*

*Figure 4. Continued*

The following figure supplements are available for figure 4:

**Figure supplement 1**. Inhibition of KPNA2 does not affect PER2 nuclear localization and cellular clock rhythmicity.

**Figure supplement 2**. Inhibition of KPNA2 does not affect CRY2 nuclear localization.

**Figure supplement 3**. Inhibition of KPNA1 (importin α5) does not significantly affect PER2 and CRY2 nuclear localization and cellular clock rhythmicity.

**Figure supplement 4**. KPNA isoforms are not critical for cellular rhythmicity.

## Inhibition of a conserved complex member, importin β, abolishes rhythms in flies

The majority of core clock components are conserved between flies (cycle, clk, per, cry) and humans (BMAL1/BMAL2, CLOCK/NPAS2, PER1/PER2/PER3, CRY1/CRY2), across 600 million years of evolution. Indeed, the kinases that regulate PER/CRY translocation to the nucleus are also conserved (dbt, sg; CKIε/CKIδ, GSK3β). Because KPNB1 was required for rhythmicity in human cells, we investigated the role of the *Drosophila* ortholog, Fs(2)Ket, *ketel*, in the fly clock by RNAi followed by locomotor activity monitoring (*Mason and Goldfarb, 2009*). We used the UAS/Gal4 system to target UAS-*ketel* RNAi to clock cells in *Drosophila* (*Brand and Perrimon, 1993*). *dicer* was also co-expressed to increase the RNA interference effect (*Dietzl et al., 2007*). Behavioral analysis of flies expressing UAS ketel RNAi in central clock cells, targeted by the relatively weak Pigment dispersing factor (Pdf)-Gal4 driver, revealed that 62% of flies exhibited no rhythms in constant darkness (DD). Although 38% of flies were rhythmic, their rhythm strength determined by fast Fourier transform (FFT) value was low (*Figure 5A–C*, *Supplementary file 1*). With a strong tim-UAS-Gal4 driver that is expressed in all clock cells, 100% of flies were arrhythmic (*Figure 5A,B*, *Supplementary file 1*). To exclude possible developmental effects on behavioral phenotypes in *ketel* knockdown flies, we restricted the *ketel* dsRNA expression to adult PDF⁺ neurons using the *pdf* Gene Switch (*pdf*-GS). We crossed *ketel* RNAi transgenic flies with *pdf*-GS and then drove *ketel* dsRNAs expression with a drug (RU486) during adulthood (*McGuire et al., 2004*). After 6–7 days in constant darkness, this conditional knockdown of *ketel* also resulted in weak to arrhythmic locomotor activity (*Figure 5D*, *Supplementary file 1*). 81% of flies were arrhythmic and the rest showed weak rhythmicity with low FFT values (*Figure 5E*, *Supplementary file 1*). These results show that importin β in PDF+ neurons is required for normal locomotor rhythms in adult flies.

To investigate whether this behavioral phenotype is related to nuclear translocation of clock proteins, we determined the cellular localization of PER in RU486 treated *pdf*-GS > *ketel* RNAi flies at ZT1, when PER is nuclear in wild type flies. PDF is expressed in the large and small ventral lateral neurons (LNvs), so we focused on PER expression in these cells. In *ketel* knockdown flies, PER levels were low and primarily cytoplasmic in large LNvs, while PER staining was intensely nuclear in control flies (*Figure 5F,G*). The small LNvs were undetectable in *ketel* RNAi transgenic flies, as is also the case in other arrhythmic fly mutants such as Clk and cyc (*Figure 5F*, right panel images) (*Park et al., 2000*). Taken together, these data suggest that like mammalian importin β, Ketel mediates the nuclear localization of PER, and consequently, is required for functioning of the core circadian clock.

## Discussion

Timely nuclear translocation of the PER/CRY (or Per/Tim) repressor complex is a critical and conserved step in maintaining rhythmicity in metazoans. However, the functional carrier molecules responsible for nuclear localization of the feedback regulators have yet to be determined. Our study provides the first evidence that KPNB1 plays a key role as a nuclear transporter of the PER/CRY complex in negative feedback repression in both mammalian and fly clocks (*Figure 6*).

Many mammalian and *Drosophila* clock studies have shown that phosphorylation and nuclear translocation of PERs and CRYs are closely coupled (*Gallego and Virshup, 2007*). We noted that CKIε, GSK3β, and DYRK2, clock kinases in both mammals and flies, are also constituents of the NRON

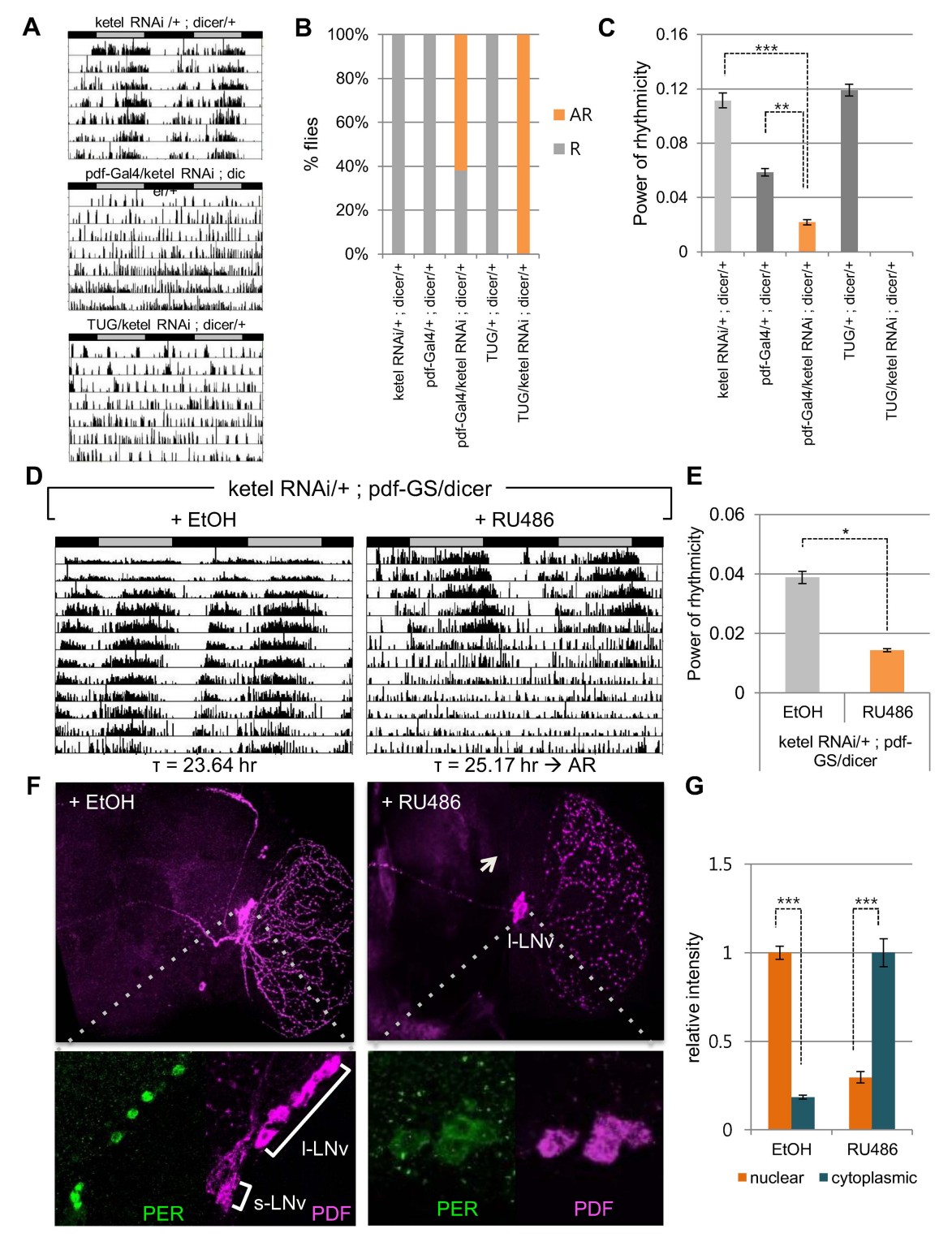

**Figure 5**. Inhibition of conserved importin β (ketel) abolishes circadian behavior in flies. (**A**) Flies in which ketel was downregulated in PDF+ cells and in all clock neurons were assayed for rest/activity rhythms in constant dark (DD). (**B**) Summary of the circadian rhythms of various flies under DD. Quantification shows percentage of rhythmic (R, orange) and arrhythmic (AR, gray) flies (n = 16–26). (**C**) Some rhythmic Pdf-Gal4/+; dicer/+ flies showed weak rhythmicity with lower fast Fourier transform value compared to both ketel RNAi/+; dicer/+ and *Pdf*-GAL4/+; dicer/+ control flies (**p < 0.001, ***p < 0.0001, by Student's *t*-test) (See also *Supplementary file 1*). (**D**) *ketel* knockdown in PDF+ cells during adulthoods leads to a long period, which eventually degenerates into arrhythmia in DD. They were fed either 500 mM RU486 or ethanol (EtOH, vehicle control) from the time of entrainment. (**E**) Rhythm

*Figure 5. continued on next page*

*Figure 5. Continued*

strength of rhythmic RU486-treated flies was lower than that of EtOH-treated control flies after 7 days in DD (\*p < 0.05, by Student's *t*-test) (See also ***Supplementary file 1***). (**F, G**) PDF expression in *ketel* knockdown flies was only detected in large lateral ventral neurons (l-LNvs), not in the small lateral ventral neurons (s-LNvs). Downregulation of KETEL in PDF[+] cells impairs nuclear translocation of PER in l-LNvs. In RU486-treated flies, PER expression in l-LNvs was detected in cytoplasm at ZT1, a time when PER is nuclear in control flies (\*\*\*p < 0.0001, by Student's *t*-test).

complex, which regulates cytoplasmic to nuclear translocation of NFAT and consequently its function in the immune systems (***Willingham et al., 2005***). The NRON complex also contains several molecules involved in protein stability and, critical for this study, nuclear translocation (KPNB1, TNPO1, CSEL1). Subsequent study showed that KPNB1 is the major transport receptor for NFAT nuclear entry and activity upon its dephosphorylation (***Sharma et al., 2011***).

Thus, we investigated the role of KPNB1 on nuclear localization of PER/CRY complex (***Figure 1***, ***Figure 1—figure supplement 1***). We found that KPNB1 knockdown trapped PER and CRY proteins at the nuclear membrane, preventing their nuclear accumulation. Consistent with this, in our transcriptional study, we found KPNB1 knockdown significantly relieved *PER1* transcriptional repression by the PER/CRY complex. Depletion of the other NRON complex translocation factors, TNPO1 and CSE1L, did not (***Figure 3***, ***Figure 3—figure supplement 1***), nor did they interact with PER or CRY proteins (data not shown). These data suggest a specific role of KPNB1 as a nuclear transport receptor directly involved in negative feedback regulation of the core clock, in addition to its previously described role in NFAT signaling.

Despite not playing a role in PER/CRY function, both TNPO1 and CSE1L displayed circadian gene expression in various mouse tissues, and also affected circadian period length (TNPO1) or were required for cellular rhythms (CSE1L) when silenced ([***Pizarro et al., 2013***], data not shown). Interestingly, the recent in situ hybridization analysis showed that TNPO1 exhibits circadian

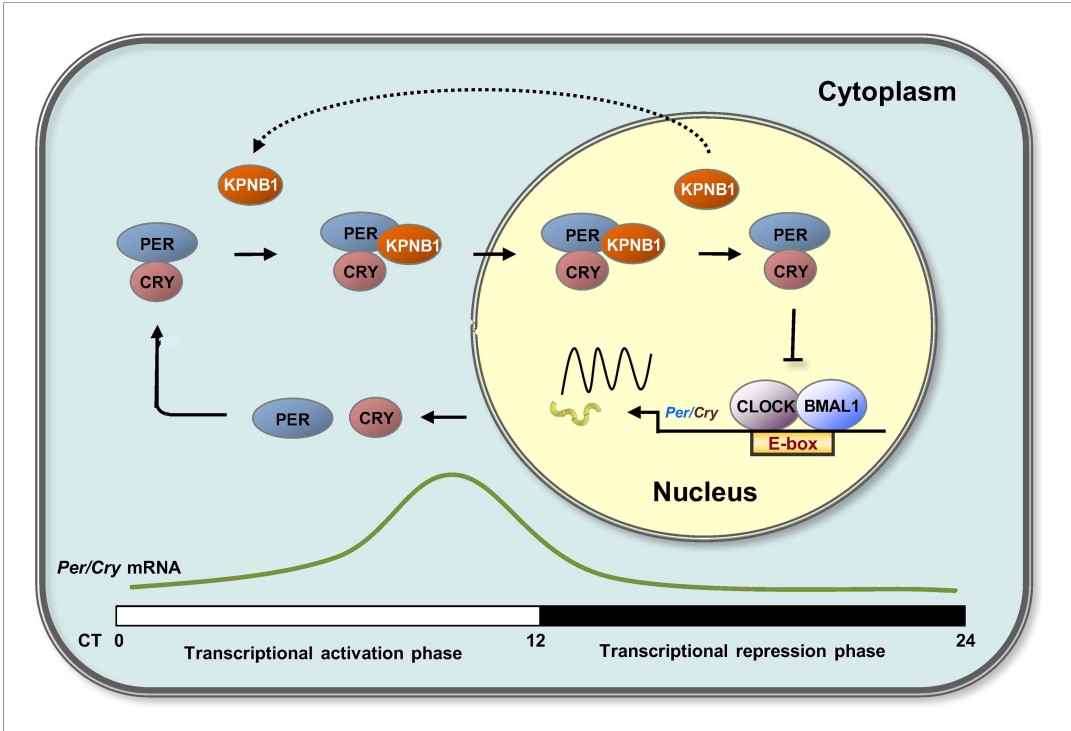

**Figure 6**. Proposed model for KPNB1 function in nuclear translocation of PERs/CRYs controlling negative feedback regulation of the molecular clock. KPNB1 directly associates with PERs/CRYs and guides their nuclear entry thus facilitating negative feedback repression of CLOCK/BMAL1-mediated transcription of the repressor genes.

expression profiles in its mRNA in SCN (*Sato et al., 2011*). Thus, these proteins may also play a role in mammalian clock function independent from PER/CRY, perhaps regulating import and export of other clock related components.

Notably, in our biochemical analyses, KPNB1 strongly interacted with PER proteins and exhibited rhythmic nucleocytoplasmic localization and concurrent nuclear accumulation with PER/CRY in transcriptional repression phase (*Figure 2*). These data suggest KPNB1 drives circadian nuclear entry of PER/CRY. This raises the question: what causes phase-dependent KPNB1 nuclear entry? Most recently, it was reported that intracellular calcium levels regulate importin β-dependent nuclear import (*Kaur et al., 2014*). Further, a previous model suggests that KPNB1 nuclear-translocates NFAT immediately after stimulation dependent intracellular calcium influx (*Sharma et al., 2011*). In this regard, several studies have demonstrated the important function of $Ca^{2+}$ dynamics in circadian clock regulation. For example, periodic $Ca^{2+}$ influx in the cytosol, resulting from circadian variation in membrane potential or by the release of Ca(2+) from ryanodine-sensitive stores in SCN neurons, is a critical process for circadian pacemaker function (*Ikeda et al., 2003*; *Lundkvist et al., 2005*). Based on this evidence, we postulate that cytosolic $Ca^{2+}$ rhythms contribute to driving timely nuclear localization of KPNB1 and its PER/CRY cargo as a critical step in negative feedback regulation of the clock.

In early models, KPNB1 mediates docking of the importin α/substrate complex in the nuclear pore. More recent evidence shows that KPNB1 can function as nuclear transport receptor independently of importin α by directly recognizing substrates such as CREB, AP-1 transcription factors, and parathyroid hormone-related protein (*Lam et al., 1999*; *Forwood et al., 2001*; *Ghildyal et al., 2005*). Interestingly, a previous report showed that importin α isoforms (RCH, QIP-1, NPI-2) mediate nuclear localization of CRY2 through direct interactions but did not investigate the functional consequence of this interaction (*Sakakida et al., 2005*). In this regard, our data suggest that in mammals KPNB1 mediates nuclear localization of the PER/CRY complex independently of importin α, as overexpression of a KPNB1 mutant lacking an importin α interaction domain had a similar phenotype (*Figure 1*, *Figure 3—figure supplement 2*). Consistent with this, both KPNA2 and KPNA1, the well-known KPNB1 interacting importin alpha proteins (importin α1, importin α5), when depleted in cells, did not affect circadian rhythmicity or subcellular localization of PER2 or CRY2 protein (*Figure 4—figure supplement 1*, *Figure 4—figure supplement 2*, *Figure 4—figure supplement 3*). In a recent report, cytoplasmic retention of PER proteins was accelerated by *KPNA2* overexpression in mouse stem cells (*Umemura et al., 2014*). However, intriguingly we found that *KPNA2* overexpression substantially reduced cells expressing PER2-Venus protein in our imaging studies (*Figure 4—figure supplement 1E*). Moreover, both the endogenous and overexpressed KPNA2 protein localized predominantly in the nucleus, which was confirmed by nuclear cytoplasmic fraction Western analysis (*Figure 4—figure supplement 1C,E*, *Figure 4—figure supplement 2A*). This suggests that KPNA2 might regulate PER2 protein level through an unknown mechanism in U2 OS cells. Thus, we speculate that these different results are likely due to the different cell types and species used.

On the other hand, knockdown of KPNB1 had strong (PER1, PER2), modest (CRY1), or no effect (CRY2) on single tagged PER and CRY proteins, while it had strong effects on all combinations of PER/CRY complexes (*Figure 1A–C*). Similarly, knockdown of KPNB1 increased cytoplasmic retention of endogenous PER1, PER2, CRY1, and CRY2 to varying degrees, probably through complex interactions (*Figure 1D*, *Figure1—figure supplement 2*). The immunoprecipitation analysis showed that KPNB1 interacted strongly with PERs compared with CRYs (*Figure 2—figure supplement 1A*). These data suggest that the PERs mediate interaction with KPNB1 and gate nuclear translocation of PER/CRY complex. This is reminiscent of a previous report indicating that the nuclear entry of CRY is mainly dependent on PER, in vivo (*Lee et al., 2001*). Finally, knockdown of KPNB1 or its fly ortholog *ketel* resulted in arrhythmic U2 OS cells or fly locomotor activity rhythms, while knockdown of most of KPNA isoforms had no significant effect on the rhythm (*Figure 4*, *Figure 5*, *Figure 4—figure supplement 4*). These data suggest a model in which KPNB1 plays a necessary role in mediating nuclear entry of the PER/CRY complex in core clock regulation (*Figure 6*).

Recently, importin β emerged as a potential molecular target for cancer prevention (*van der Watt et al., 2013*). So far, several novel small molecule regulators of KPNB1 have been identified, though further investigation is needed to improve their specificity and efficacy (*Hintersteiner et al., 2010*; *Soderholm et al., 2011*). In this respect, our study provides a rationale for KPNB1 as molecular target for regulating clock physiology.

# Materials and methods

## Cell culture and reagents

U2 OS or HEK293T cells were cultured in Dulbecco's modified Eagle's medium supplemented with 10% FBS, 1% L-Glutamine, and 1% penicillin-streptomycin (Invitrogen, Grand Island, NY, United States) at 37°C under 5% $CO_2$. The cells were transfected with siRNAs using Lipofectamine RNAiMAX (Invitrogen) and DNA plasmids using Fugene HD reagents (Promega, Madison, WI, United States). The combined transfection of DNA plasmids with siRNA into the cells were performed using Lipofectamine 2000 (Invitrogen).

## Plasmids

For overexpression of NRON in cells and generation of its transgenic cell lines, full-length genomic DNA fragments of NRON from U2 OS cells was obtained by PCR with primers containing the flanking restriction sites (NotI, XhoI) and inserted into pcDNA 3.1 mammalian expression vector (Invitrogen). Plasmids encoding several flag-tagged clock proteins (Flag-PER1, Flag-PER2, Flag-CRY1, Flag-CRY2, Flag-CLOCK, Flag-REVERBα, Flag-CHRONO) were generated as described previously (*Anafi et al., 2014*). For plasmids expressing Venus-tagged PER1, PER2, CRY1, CRY2 (PER1-Venus, PER2-Venus, CRY1-Venus, CRY2-Venus) or BiFC plasmids expressing Venus N terminal (VN) or C-terminal (VC)-tagged PER1, PER2, CRY1, and CRY2 (PER1-VN, PER2-VN, CRY1-VC, CRY2-VC), full-length DNA fragment of the each of genes was subcloned into pCMV-Venus, pCMV-VN, or pCMV-VC using a restriction-free cloning method as described previously (*Shyu et al., 2006*; *Bond and Naus, 2012*). For luciferase assays, pCMV Sport6 plasmids encoding full length cDNAs of CLOCK, BMAL1, PER1, PER2, CRY1, CRY2, KPNB1 was obtained from the Mammalian Gene Collection (Thermo Fisher Scientific, Grand Island, NY, United States). For generation of plasmids encoding mutant KPNB1, DNA fragments of the KPNB1 gene with an importin-α binding domain deletion encoding a truncated protein (1–771) were sub-cloned into pCMV Sport6 expression vector (Invitrogen) using NotI and SalI restriction enzymes.

## Luciferase assay

HEK293T cells transfected with plasmids encoding proteins as indicated in the figures were assayed for luciferase reporter activity using DualGlo luciferin reagent (Promega) 24 hr post-transfection according to the manufacturer's protocol.

## Antibodies

Anti-PER2 used for immunoblot analysis of immunoprecipitated samples from mouse liver extracts were kindly provided by Dr Cheng Chi Lee's lab (University of Texas Health Science Center at Houston). Anti-KPNB1 (A300-482A) for immunoblotting analysis was obtained from Bethyl Labratories (Montgomery, TX, United States), and Anti-KPNB1 (ab2811) for immunoprecipiation with mouse liver extracts or immunostaining of U2 OS cells was purchased from Abcam (Cambridge, MA, United States). For ChIP or immunostaining experiments, Anti-PER1 (AB2201, Millipore, Darmstadt, German) and Anti-PER2 (NB100-125, Norvus Biologicals, Littleton, CO, United States) were used. Anti-Flag (F7425, Sigma, St. Louis, MO), Anti-GFP (G1544, Sigma), Anti-PER1 (AB2201, Millipore), Anti-PER2 (20359-1-AP, Proteintech, Chicago, IL, United States), Anti-CRY1 (sc-33177, Santa Cruz Biotech, Dallas, Texas, United States), Anti-CRY2 (13997-1-AP, Proteintech), were used for immunopreciptation, immunoblotting, and immunostaining experiments as indicated in figures. Normal mouse IgG (NI03) was purchased from Calbiochem. Anti-Tubulin (Ab7291, Abcam), anti-TBP (22006-1-AP, Proteintech), anti-Lamin (#2032, Cell Signaling, Danvers, MA, United States). and anti-GAPDH (sc25778, Santa Cruz Biotech) were used as loading control antibodies for cytoplasmic, nuclear, and total protein extracts respectively.

## Cytosol-nuclear fractionation

U2 OS cells were co-transfected with each of the following DNA plasmids encoding Venus (V)-tagged PER1, PER2, CRY1, or CRY2 (PER1-V, PER2-V, CRY1-V, CRY2-V) and siRNA against KPNB1. Cells were collected 48 hr post-transfection by briefly trypsinizing and washing twice in ice cold PBS. Nuclear–cytoplasmic fractionation of the cells was performed using the NE-PER Nuclear and Cytoplasmic Extraction Reagents kit (78835, Thermo Fisher Scientific) according to the manufacturer's protocol. The same kit protocol was used for preparation of cytoplasmic and nuclear extracts from mouse liver tissues collected at 4 hr interval for 24 hr in constant darkness.

## Co-immunoprecipitation and immunoblotting

At 48 hr post-transfection of plasmids encoding PER2-Venus into U2 OS cells, the cell lysates were harvested in radioimmunoprecipitation assay (RIPA) buffer (50 mM HEPES [pH 7.4], 150 mM NaCl, 1% NP-40, 1 mM EDTA, 1 mM EGTA, 1 mM phenylmethylsulfonyl fluoride, 0.5% sodium deoxycholate, 1 mM NaF, 1 mM $Na_3VO_4$, and protease inhibitor cocktail [Roche, Indianapolis, IN, United States]). Protein-G coated magnetic beads (10004D, Life Technologies, Grand Island, NY, United States) were pre-incubated with 3 µg of anti-GFP (G1544, Sigma) antibody at 4°C for 6 hr. The antibody conjugated beads were incubated with equal amounts of total protein at 4°C overnight. The final immune complexes were analyzed by immunoblot assay using antibodies as described. For tissue immunoprecipitation, liver tissue was homogenized in 1 vol of RIPA buffer 1 (50 mM Tris-HCl [pH 8.0], 450 mM NaCl, 1% Triton X-100, 1 mM EDTA, 1 mM EGTA, 1 mM phenylmethylsulfonyl fluoride, 0.5% sodium deoxycholate, 1 mM NaF, 1 mM $Na_3VO_4$, and 1 protease inhibitor cocktail [Roche]). Homogenates were cleared by dilution with 2 vol of RIPA buffer 2 (RIPA buffer 1 without NaCl). Further procedures were as described above. Immunoblot analyses were performed on 7.5% sodium dodecyl sulfatepolyacrylamide gels and transferred to polyvinylidene difluoride membranes (Immobilon P; Millipore). Target proteins were detected with antibodies as indicated. The immune complexes were visualized with an ECL detection kit (Pierce, Grand Island, NY, United States).

## ChIP

For ChIP analysis, cell lysates were collected from U2 OS cells and the following 'Materials and methods' were performed as described previously (*Schmidt et al., 2009*). Briefly, the pre-cleared chromatin was immunoprecipitated overnight at 4°C by agitating with 5 µg of PER1 or PER2 antibody as described. The cell extracts incubated in the absence of antibody were used for input controls. Immune complexes were collected by incubation with protein-G coated magnetic beads (10004D, Life Technologies) and the final eluted DNA was extracted by phenol–chloroform–isoamyl alcohol (25:24:1) and ethanol precipitation. The primer sets used for ChIP Quantitative PCR (qPCR) analysis of human *Per1* promoter region spanning canonical (CACGTG) were as follows: Forward primer: 5′-TCTCCCTCTCTCCTCCCTTCC-3′, Reverse primer: 5′-GCCTGATTGGCTAGTGGTCTT-3′, Probe: GTGTGTGACACAGCCCTGACC.

## BiFC and IF assays

At 24 hr post-transfection of expression vectors encoding Venus-tagged clock proteins (PER1-Venus, PER2-Venus, CRY1-Venus, CRY2-Venus) or BiFC plasmids expressing (PER1-VN, PER2-VN, CRY1-VC, CRY2-VC) as indicated, the cells were fixed with 4% paraformaldehyde in PBS and visualized using GFP filter set in fluorescence microscopy. For IF analysis, U2 OS cells was incubated with various antibodies as indicated by secondary antibodies conjugated to Alexa Fluor 488 and/or 568 (Invitrogen). Cells were visualized using FITC/TRITC/DAPI filter set in fluorescence microscopy.

## Bioluminescence recording and data analysis

Real-time bioluminescence of p*Per2* or p*Bmal1* dLuc reporter U2 OS cells after synchronization with 1 µM Dex (Sigma) were monitored using a LumiCycle luminometer (Actimetrics, Wilmette, IL, United States) as previously described (*Zhang et al., 2009*). The period of the luminescence data was determined through the WaveClock algorithm (*Price et al., 2008*).

## Reverse transcription and quantitative PCR

cDNAs were generated with the High Capacity cDNA Archive Kit using the manufacturer's protocol (Applied Biosystems, Grand Island, NY, United States). qPCR reactions were conducted using iTaq PCR mastermix (BioRad, Hercules, CA, United States) in combination with gene expression assays (Applied Biosystems) on a 7800HT Taqman machine (Applied Biosystems). *GAPDH* was used as an endogenous control for all experiments.

## Whole-mount brain immunocytochemistry

3–5 days old flies were collected at indicated ZT on the fourth day of LD entrainment. Fly heads of each genotype were opened and brains were immediately fixed with 4% paraformaldehyde (in 1× PBS) for 15–20 min, followed by dissection in 1× PBST (0.3% Triton X-100 in PBS) at room temperature. After 30min-wash with 1× PBST at room temperature, brains were blocked with 5% normal donkey serum (NSD)

for 20 min, and then incubated overnight at 4°C with primary antibody: rat anti-PER (UPR34, 1:1000) and mouse anti-PDF (C7, Developmental Studies Hybridoma Bank, University of Iowa, 1:500). After 30-min wash in 1× PBST at room temperature, brains were incubated with secondary antibodies (Jackson ImmunoResearch Laboratories, West Grove, PA, United States, 1:500) for 2 hr in NDS at room temperature, followed by extensive 30 min-wash. Samples were imaged using Leica TCS SP5 confocal microscope. 8–10 brains were examined at each time point.

## Fly strains and behavioral assay

Targeted expressions of *ketel* RNAi in PDF+ cells and tim-expressing clock neurons were performed using the UAS/GAL4 system. We tested two independent UAS-*ketel* RNAi lines (v22348, v107622; Vienna Drosophila RNAi center) and isogenic w[1118] (iso[31]) strain was used as a wild-type control. Male flies of the UAS-*ketel* RNAi lines or iso[31] (control) were crossed to female flies from Gal4 drivers. Male progenies of the crosses were entrained to a 12 hr:12 hr LD cycle at 25°C at least for 3 days. Individual flies were loaded in the glass locomotor tubes containing 5% sucrose and 2% agar. Locomotor activity was measured for 7–15 days in DD using the *Drosophila Activity Monitoring* System as previously described (*Williams et al., 2001*). Power of rhythmicity was determined by FFT value. Flies with FFT value >0.01 were counted as a rhythmic. This activity records were analyzed using ClockLab software (Actimetrics).

## Acknowledgements

We thank Ted Abel for helpful conversations. This work is supported by the National Institute of Neurological Disorders and Stroke (1R01NS054794-06 to JBH), the Defense Advanced Research Projects Agency (DARPA-D12AP00025, to John Harer, Duke University), and by the Penn Genome Frontiers Institute under a HRFF grant with the Pennsylvania Department of Health.

# Additional information

### Funding

| Funder | Grant reference | Author |
|---|---|---|
| National Institute of Neurological Disorders and Stroke (NINDS) | JBH, 1R01NS054794-06 | John B Hogenesch |
| Defense Advanced Research Projects Agency (DARPA) | John Harer, Duke University, DARPA-D12AP00025 | John B Hogenesch |
| Pennsylvania Department of Health | Penn Genome Frontiers Institute under a HRFF | John B Hogenesch |

The funders had no role in study design, data collection and interpretation, or the decision to submit the work for publication.

### Author contributions

YL, JBH, Conception and design, Acquisition of data, Analysis and interpretation of data, Drafting or revising the article, Contributed unpublished essential data or reagents; ARJ, Conception and design, Acquisition of data, Analysis and interpretation of data, Contributed unpublished essential data or reagents; LJF, Acquisition of data, Analysis and interpretation of data, Contributed unpublished essential data or reagents; AS, Conception and design, Analysis and interpretation of data, Contributed unpublished essential data or reagents

### Author ORCIDs

Yool Lee, http://orcid.org/0000-0003-0498-3342
A Reum Jang, http://orcid.org/0000-0001-8347-7848
Lauren J Francey, http://orcid.org/0000-0002-9700-2277
Amita Sehgal, http://orcid.org/0000-0001-7354-9641
John B Hogenesch, http://orcid.org/0000-0002-9138-8973

## Additional files

**Supplementary file**
• Supplementary file 1. Effects of downregulating ketel genes on free-running circadian locomotor rhythms.

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
