## [Decision Letter]

Thank you for submitting your work entitled “KPNB1 mediates PER/CRY nuclear translocation and circadian clock function” for peer review at *eLife*. Your submission has been favorably evaluated by a Senior Editor and three reviewers, one of whom served as Guest Reviewing Editor.

The reviewers have discussed the reviews with one another and the Reviewing Editor has drafted this decision to help you prepare a revised submission.

The study by Lee et al. describes the importance for KPNB1, a member of the importin β family, for nuclear localization of clock proteins, in particular PERIOD proteins in both human cells as well as in *Drosophila*. Timed nuclear accumulation of members of the negative limb within the circadian clockwork has long been known to be critical for a functional circadian clock. Circadian rhythms of nuclear appearance of clock proteins as well as functional nuclear localization and nuclear export sequences (NLS and NES) in these proteins have been reported. The dynamics of this is important since it ensures the correct timing of negative feedback repression of clock genes. Here, the authors show that KPNB1 depletion results in primarily cytoplasmic localization of PER in human cells and *Drosophila*. Timely subcellular localization of PER2, CRY1, and KPNB1 coincides well and KPNB1 mainly interacts with PER1/2. Thus, the authors concluded that KPNB1 controls nuclear translocation of PER/CRY complex via direct interaction with PER. Intriguingly, KPNB1 mediated nuclear translocation of PER/CRY complex is not dependent on KPNA2 (importin α1) which is a well-known partner of importin β proteins and identified as major carrier protein for PER/TIM repressor complex in flies.

Essential revisions:

1) Clarify the role of KPNB1 knockdown on clock gene expression (see Reviewer#1, point 1).

2) Is KPNB1 really required for CIRCADIAN feedback repression (see Reviewer#1, point 4)?

3) Better define (and discuss) the roles of KPNA family members (see Reviewer #2, point 1 and Reviewer #3).

Reviewer #1:

The study by Lee et al. describes the importance for KPNB1 (importin β) for nuclear localization of clock proteins, in particular PERIOD proteins in both human cells as well as in *Drosophila*. Timed nuclear accumulation of members of the negative limb within the circadian clockwork has long been known to be critical for a functional circadian clock. Circadian rhythms of nuclear appearance of PERs, CRYs as well as TIM (in flys) as well as functional nuclear localization and nuclear export sequences (NLS and NES) in these proteins have been reported. The dynamics of this is important since it ensures the correct timing of negative feedback repression of clock genes.

Here, the authors show that KPNB1 (but not KPNA2 – importin α) depletion in human cells results in primarily cytoplasmic localization of PER proteins (and to a lesser extent also of CRYs) both for overexpressed as well as for endogenous proteins (a nice and complete set of experiments). In addition, PER proteins directly bind to KPNB1 (are the NLS required?). Also very convincing and carefully conducted are the results they found in *Drosophila*: again, PER is shifted towards more cytosolic localization upon depletion of *ketel*, the *Drosophila* ortholog of KPNB1. Probably as a consequence, flies with KPNB1 depletion in pdf cells or all clock cells are largely arrhythmic.

Less convincing are the experiments aiming to demonstrate effects on the circadian clockwork in mammalian cells (see below).

Overall, however, the manuscript is of high quality and should eventually be published, if the following points can be convincingly addressed:

1) The effects of KPNB1 knockdown on clock gene expression show conflicting results: either upregulation of PER1 pre-mRNA and downregulation of *BMAL1* pre-mRNA (Figure 3) or the opposite (Figure 4). In addition, the effects in Figure 3 are very subtle (1.2-fold upregulation of CRY1 pre-mRNA - I don't believe that there is statistical significance given that the (partly high) error bars are SEM with n=3). What about endogenous clock gene levels upon overexpression of KPNB1? Maybe this helps to solve the issue.

2) In addition, the effects on circadian rhythmicity should be better characterized: are there period effects? Why does the *Per2*-reporter show a low and the *Bmal1*-reporter a high magnitude, if KPNB1 knockdown is supposed to lead to less inhibition of E-boxes?

3) Figure 4: controls do oscillate with a 16-18 hour period (in conflict with Figure 4 and in conflict with knockdown time series). This is odd and could be normalization artefact - at least the authors should show *BMAL1* as an antiphasic control.

4) Figure 2: The authors claim that KPNB1 is responsible for circadian accumulation and feedback repression, hence they only observe correlations. To unambiguously show that KPNB1 is required for circadian feedback repression, they should analyze circadian protein accumulation (especially CRYs, since they are the major repressors and still go in the nucleus without KPNB1) in wild-type and KPNB1 depleted cells together with analysis of clock gene pre-mRNA over time.

Reviewer #2:

In this study, the authors explored the molecular mechanism for nuclear translocation of circadian repressor complex, PER/CRY. They identified KPNB1, a member of the importin β family, plays major role for nuclear translocation of PER1/2, PER/CRY complex and hence feedback repression. Timely subcellular localization of PER2, CRY1, and KPNB1 coincides well and KPNB1 mainly interacts with PER1/2. Thus, the authors concluded that KPNB1 controls nuclear translocation of PER/CRY complex via direct interaction with PER. Intriguingly, KPNB1 mediated nuclear translocation of PER/CRY complex is not dependent on KPNA2 (importin α1) which is a well-known partner of importin β proteins and identified as major carrier protein for PER/TIM repressor complex in flies. Overall, the experiments are well executed and this finding will enhance our understanding of detailed biochemical mechanisms underlying timely controlled nuclear translocation of circadian repressor complex.

Although the results demonstrating KPNB1 is the major carrier protein for PERs are very solid, the conclusion about KPNB1 being the major carrier protein for PER/CRY complex in mammals and PER/TIM complex in flies needs to be further qualified (see below).

1) As cited in the text, [28] demonstrated that importin α family members (including importin α1, KPNA2) control nuclear translocation of CRY2 but not importin β. The data from Sakakida et al., is consistent with this study showing that depletion of KPNB1 produces minimal effects on CRY2 translocation and partial effects on CRY1 translocation (Figure 1). Based on these two studies, this reviewer thinks that Importin α1 might control translocation of mainly CRY2 and some of CRY1. This is somewhat similar to the case of flies, where translocation of TIM (fly counterpart of CRY at least in the functional aspect) is controlled by importin α1 (Jang AR et al., Plos Genetics 2014, *Drosophila* TIM binds importin α1, and acts as an adapter to transport PER to the nucleus). Figure 3 luciferase results indicated that CRY1/CRY2 represses even better than PER1/PER2. Then, confusion comes from the data showing KPNA1 deficiency did not produce any defects in cellular rhythms. Thus, this reviewer is wondering how CRY2 localized in KPNA1 knockdown cells. By performing this experiment we might be able to clarify the reason why KPNA1 depletion did not produce any effects on cellular rhythms. Is that due to either (2) KPNA1 does not playing any role in translocation of clock proteins (mainly CRY2) on the contrary to previous report and different from the fly case (or there might be cell-type specific issue?) or (27) nuclear translocation defects of CRY2 not producing any effects on feedback repression suggesting that main repressor of feedback circuitry is PER not CRY?

2) The authors showed that knockdown of *ketel* (fly homolog of KPNB1) in clock cells made flies arrhythmic via inhibiting nuclear translocation of PER, hence claiming KPNB1 mediates nuclear entry of PER. In another paper lead by some of the authors in this study, they demonstrated that importin α1 protein is critical for nuclear translocation of fly circadian repressor complex PER/TIM, through direct interaction between TIM and importin α1 (Jang AR et al., Plos Genetics, 2014). There, Jang AR et al. proposed the model in which TIM bound to importin α1 acts as a carrier for PER translocation together with importin β. Thus, this reviewer is wondering whether nuclear translocation of TIM is also inhibited in *ketel* mutant, which will be a slightly different case to the mammals. Or, very little amounts of PER translocate to the nucleus in a TIM-independent manner through *ketel*. The authors showed only the PER patterns in *ketel* knockdown flies. By showing TIM patterns in *ketel* knockdown flies, it will be clarified that PER translocate into the nucleus as a PER/TIM/importin α1/importin β complex or via two pathways one is TIM/importin α1 dependent and the other is *ketel* dependent as in mammals.

3) In the Discussion, the authors state: “Similarly, knockdown of KPNB1 resulted in cytoplasmic retention of endogenous PER1, PER2, CRY1, and CRY2 substantially (Figure 1, Figure 1—figure supplement 2).” However, I do not see any substantial effects on CRY2 protein. Rather, this data clearly indicated KPNB1 did not play any roles in nuclear translocation of CRY2 protein. Thus, this statement needs to be rewritten.

Reviewer #3:

In the manuscript entitled “KPNB1 mediates PER/CRY nuclear translocation and circadian clock function” by Lee et al., the authors investigate the role of KPNB1 (Importin β subunit) for nuclear translocation of PER proteins and its functional implication for circadian clock oscillation in mammalian cells and flies. The authors clearly show that KPNB1 critically regulates the nuclear translocation of PER proteins which are largely independent from CRY proteins in U2OS cells. Interestingly, a knock-down of KPNB1 blocked the nuclear translocation of the PER/CRY complex, suggesting that PER proteins played a dominant role in the nuclear translocation of the PER/CRY complex in mammalian cells. In addition, the authors exhibit that the loss of KPNA2 does not block the nuclear translocation of PER proteins and circadian molecular rhythms in mammalian cells. Moreover, the authors show that a decrease in *Drosophila* Importin β also blocks the nuclear translocation of PER proteins and results in the severe impairment of circadian locomotor activity rhythms in flies. By gathering these findings, the authors conclude that KPNB1 is the essential factor to elicit the cyclic nuclear translocation of PER/CRY in the negative-feedback regulation of the circadian clock.

In general, the study includes novel findings, especially for the regulation mechanism of subcellular PER/CRY dynamics, which is essential for the circadian clock. However, the reviewer requires the authors discuss the following points before publication.

Different from KPNA1, expression level of KPNA2 is usually very low in normal somatic cells. On the other hand, KPNA2 highly expressed in early embryo or pluripotent stem cells as well as in some cancer cells. For example, the expression level of the Kpna2 transcript is extremely low in adult mice liver (Koike et al., Science, 2012). In addition, it has recently been reported that KPNA2 accelerates cytoplasmic localization of proteins in some cases (e.g. Yasuhara et al., Mol. Cell, 2012; Alshareeda et al., Br. J. Cancer, 2015). Through the accumulated findings, KPNA2 has been thought to be a unique protein among the KPNAs, and does not represent the function of other KPNAs in somatic cells. Thus, the reviewer feels curious as to why the authors performed knock-down study against KPNA2 instead of KPNA1 or any other KPNAs. The authors might have taken into account a recently reported study which mentioned KPNA2 as a factor to obstruct circadian clock cycling in mouse embryonic stem (ES) cells (Umemura et al., PNAS, 2014). Umemura et al. showed the high level expression of KPNA2 and cytoplasmic retention of PER protein in undifferentiated ES cells. In addition, overexpression of KPNA2 did not facilitate nuclear translocation, rather accelerated cytoplasmic retention of PER proteins. Strikingly, the authors here clearly showed that knock-down of KPNA2 in U2OS cells resulted in the exclusive nuclear localization of PER2 and increased amplitude of *Per2*:dluc circadian rhythms (Figure 4—figure supplement 1). These results are all well compatible with the results mentioned in the paper by Umemura et al. Therefore, the reviewer recommends that the authors discuss this point.

---

## [Author Response]

*1) Clarify the role of KPNB1 knockdown on clock gene expression (see Reviewer#1, point 1)*.

*2) Is KPNB1 really required for CIRCADIAN feedback repression (see Reviewer#1*, *point 4)?*

*3) Better define (and discuss) the roles of KPNA family members (see Reviewer #2, point 1 and Reviewer #3)*.

Reviewer #1:

*1) The effects of KPNB1 knockdown on clock gene expression show conflicting results: either upregulation of PER1 pre-mRNA and downregulation of* BMAL1 *pre-mRNA (*Figure 3*) or the opposite (*Figure 4*). In addition, the effects in*
Figure 3
*are very subtle (1.2-fold upregulation of CRY1 pre-mRNA - I don't believe that there is statistical significance given that the (partly high) error bars are SEM with n=3). What about endogenous clock gene levels upon overexpression of KPNB1? Maybe this helps to solve the issue*.

We agree and did the experiment you suggested. In parallel, we tested both KPNB1 knockdown and overexpression on endogenous clock components regulated primarily by the E-box loop. This showed that KPNB1 knockdown led to significant up-regulation of most of the E-box-controlled genes, PER1, CRY1, DBP, REV-ERBβ. Conversely, these genes are repressed by KPBN1 over-expression (Figure 3, subsection “KPNB1 regulates transcriptional repressor activity of PERs/CRYs and circadian gene expression independently of importin α (KPNA2)”, and Figure 3, legend). Thus, we believe these results provide further evidence that KPNB1 directly mediates PER/CRY repressor activity for regulating E-box dependent clock gene expression.

*2) In addition, the effects on circadian rhythmicity should be better characterized: are there period effects? Why does the* Per2*-reporter show a low and the* Bmal1*-reporter a high magnitude, if KPNB1 knockdown is supposed to lead to less inhibition of E-boxes?*

Yes, the period length of both the *Per2* and *Bmal1* reporters get longer and dampen more quickly, as representatively shown in the detrended graph (Figure 4 right panel). The *Per2* reporter we use is longer and contains E-boxes, D-boxes, and ROR elements, making interpretation of its direction difficult to know a priori.

*3)*
Figure 4*: controls do oscillate with a 16-18 hour period (in conflict with*
Figure 4
*and in conflict with knockdown time series). This is odd and could be normalization artefact - at least the authors should show* BMAL1 *as an antiphasic control*.

Per your suggestion, we re-did this experiment in the revised Figure 4. PER1 and PER2 mRNA expressions showed anti-phasic pattern with *BMAL1*. Levels of *Per1* go up, while levels of *Bmal1* are unchanged, consistent with Figure 3. *Bmal1* is also antiphase to *Per1* and *Per2*.

*4)*
Figure 2*: The authors claim that KPNB1 is responsible for circadian accumulation and feedback repression, hence they only observe correlations. To unambiguously show that KPNB1 is required for circadian feedback repression, they should analyze circadian protein accumulation (especially CRYs, since they are the major repressors and still go in the nucleus without KPNB1) in wild-type and KPNB1 depleted cells together with analysis of clock gene pre-mRNA over time*.

We agree and did a time course of control and KPNB1 knockdown cells for both cytoplasmic and nuclear fractions of clock proteins and pre-mRNAs levels. As shown in Figure 4, KPNB1 knockdown markedly reduced or altered the rhythmic accumulation of PER2, CRY1, and CRY2 (Figure 4, legend). Correspondingly, the mRNA levels of several clock genes were arrhythmic in KPNB1-depleted cells (Figure 4). Thus, these data further strengthen the conclusion that KPNB1 is required for circadian feedback repression of clock gene expression.

Reviewer #2:

*1) As cited in the text,*
[28]
*demonstrated that importin α family members (including importin α1, KPNA2) control nuclear translocation of CRY2 but not importin β. The data from Sakakida et al., is consistent with this study showing that depletion of KPNB1 produces minimal effects on CRY2 translocation and partial effects on CRY1 translocation (*Figure 1*). Based on these two studies, this reviewer thinks that Importin α1 might control translocation of mainly CRY2 and some of CRY1. This is somewhat similar to the case of flies, where translocation of TIM (fly counterpart of CRY at least in the functional aspect) is controlled by importin α1 (Jang AR et al., Plos Genetics 2014,* Drosophila *TIM binds importin α1, and acts as an adapter to transport PER to the nucleus).*
Figure 3
*luciferase results indicated that CRY1/CRY2 represses even better than PER1/PER2. Then, confusion comes from the data showing KPNA1 deficiency did not produce any defects in cellular rhythms. Thus, this reviewer is wondering how CRY2 localized in KPNA1 knockdown cells. By performing this experiment we might be able to clarify the reason why KPNA1 depletion did not produce any effects on cellular rhythms. Is that due to either (*[2]*) KPNA1 does not playing any role in translocation of clock proteins (mainly CRY2) on the contrary to previous report and different from the fly case (or there might be cell-type specific issue?) or (*[27]*) nuclear translocation defects of CRY2 not producing any effects on feedback repression suggesting that main repressor of feedback circuitry is PER not CRY?*

Thank you for your comments and suggestions. The nomenclature of the KPNA1/2 genes is confusing. In both human and mouse KPNA genes, KPNA1 is importin α5 (IPOA5), while KPNA2 is Importin α1 (IPOA1).

As you suggested, we tested KPNA2 (Importin α1) knockdown effect on subcellular localization of endogenous or ectopically expressed CRY2 localization and found no significant effects in our nuclear cytoplasmic fractionation and immunofluorescence analysis (Figure 4—figure supplement 2; Figure 4—figure supplement 2, legend). We also knocked down KPNA1 and tested its effects on rhythms in U2 OS cells driven by *Per2*::luc as well as PER2/CRY2 localization and found no significant changes (Figure 4—figure supplement 3; Figure 4—figure supplement 3 legend). In addition, in our earlier work, we knocked down KPNA1, KPNA3, KPNA4, KPNA5, KPNA6, and KPNA7 (Zhang et al., Cell, 2009), with no measurable effects on circadian rhythms in *Bmal1*::luc activity in U2 OS cells.

*2) The authors showed that knockdown of* ketel *(fly homolog of KPNB1) in clock cells made flies arrhythmic via inhibiting nuclear translocation of PER, hence claiming KPNB1 mediates nuclear entry of PER. In another paper lead by some of the authors in this study, they demonstrated that importin α1 protein is critical for nuclear translocation of fly circadian repressor complex PER/TIM, through direct interaction between TIM and importin α1 (Jang AR et al., Plos Genetics, 2014). There, Jang AR et al. proposed the model in which TIM bound to importin α1 acts as a carrier for PER translocation together with importin β. Thus, this reviewer is wondering whether nuclear translocation of TIM is also inhibited in* ketel *mutant, which will be a slightly different case to the mammals. Or, very little amounts of PER translocate to the nucleus in a TIM-independent manner through* ketel*. The authors showed only the PER patterns in* ketel *knockdown flies. By showing TIM patterns in* ketel *knockdown flies, it will be clarified that PER translocate into the nucleus as a PER/TIM/importin α1/importin β complex or via two pathways one is TIM/importinαα1 dependent and the other is* ketel *dependent as in mammals*.

We think it is very unlikely that TIM would be nuclear in *ketel* mutants. For one thing, a second *ketel*-dependent (and TIM-independent mechanism) would result in PER being partly nuclear in *ketel* mutants. Also, PER would be partly nuclear in tim mutants. Neither one is the case. While PER is unstable in the cytoplasm without TIM, it is stable in the nucleus, so nuclear PER would have been visible in tim mutants. Thus we believe that Importin alpha typically requires an importin beta, and PER likely translocates into the nucleus as a PER/TIM/importin α1/importin β complex in flies.

*3) In the Discussion, the authors state: “Similarly, knockdown of KPNB1 resulted in cytoplasmic retention of endogenous PER1, PER2, CRY1, and CRY2 substantially (*Figure 1*,*
Figure 1—figure supplement 2*).” However, I do not see any substantial effects on CRY2 protein. Rather, this data clearly indicated KPNB1 did not play any roles in nuclear translocation of CRY2 protein. Thus, this statement needs to be rewritten*.

We agree that KPNB1 does not affect nuclear translocation of CRY2 protein when ectopically expressed. In fact, if we looked at endogenous CRY2 localization carefully upon KPNB1 knockdown as shown in Figure 1 and Figure 1—figure supplement 2, we observed slightly more cytoplasmic accumulation of CRY2 proteins in the KPNB1-depleted cells than control cells, probably due to its interaction with endogenous PER proteins. This also corresponds with BiFC data showing KPNB1 knockdown effects on various combinations of PER/CRY complexes (Figure 1). Thus, we revised the sentence to: “Similarly, knockdown of KPNB1 increased cytoplasmic retention of endogenous PER1, PER2, CRY1, and CRY2 to varying degrees, probably through complex interactions” (Discussion).

Reviewer #3:

*[…] Different from KPNA1, expression level of KPNA2 is usually very low in normal somatic cells. On the other hand, KPNA2 highly expressed in early embryo or pluripotent stem cells as well as in some cancer cells. For example, the expression level of the Kpna2 transcript is extremely low in adult mice liver (Koike et al., Science, 2012). In addition, it has recently been reported that KPNA2 accelerates cytoplasmic localization of proteins in some cases (e.g. Yasuhara et al., Mol. Cell, 2012; Alshareeda et al., Br. J. Cancer, 2015). Through the accumulated findings, KPNA2 has been thought to be a unique protein among the KPNAs, and does not represent the function of other KPNAs in somatic cells. Thus, the reviewer feels curious as to why the authors performed knock-down study against KPNA2 instead of KPNA1 or any other KPNAs. The authors might have taken into account a recently reported study which mentioned KPNA2 as a factor to obstruct circadian clock cycling in mouse embryonic stem (ES) cells (Umemura et al., PNAS, 2014). Umemura et al. showed the high level expression of KPNA2 and cytoplasmic retention of PER protein in undifferentiated ES cells. In addition, overexpression of KPNA2 did not facilitate nuclear translocation, rather accelerated cytoplasmic retention of PER proteins. Strikingly, the authors here clearly showed that knock-down of KPNA2 in U2OS cells resulted in the exclusive nuclear localization of PER2 and increased amplitude of* Per2*:dluc circadian rhythms (*Figure 4—figure supplement 1*). These results are all well compatible with the results mentioned in the paper by Umemura et al. Therefore, the reviewer recommends that the authors discuss this point*.

In several previous reports (1996∼2001), KPNA2 (importin α1) was the predominant form interacting with KPNB1 in both human and mouse species (Görlich et al, EMBO J. 1996, 15(8):1810-7; Kutay et al., Cell. 1997, 90(6):1061-71; Cingolani et al., Nature. 1999, 399(6733):221-9p; Köhler et al., Mol Cell Biol. 1999, 19(11):7782-91; Catimel et al., J Biol Chem. 2001. 276(36):34189-98). Thus we chose KPNA2 (importin α1) for our representative knockdown study.

Further, as you suggested, we tested KPNA1 (Importin α5) knockdown effect on subcellular localization of endogenous or ectopically expressed PER2 or CRY2 localization and found no significant effects in fluorescence analysis (Figure 4—figure supplement 3; Figure 4—figure supplement 3, legend). We also knocked down KPNA1 and tested its effects on rhythms in U2 OS cells driven by *Per2*::luc (Figure 4—figure supplement 3; Figure 4—figure supplement 3, legend) and found no significant changes. In addition, in our earlier work, we knocked down KPNA1, KPNA3, KPNA4, KPNA5, KPNA6, and KPNA7 (Zhang et al., Cell, 2009), with no measurable effects on circadian rhythms in *Bmal1*::luc activity in U2 OS cells.

Regarding the effect of KPNA2 overexpression on PER2 localization (Umemura et al.), intriguingly, we found that KPNA2 overexpression substantially reduced cells expressing PER2-Venus protein in our analysis (Figure 4—figure supplement 1; Figure 4—figure supplement 1, legend). Moreover, both the endogenous and overexpressed KPNA2 protein was localized predominantly in the nucleus (Figure 4—figure supplement 1; Figure 4—figure supplement 1, legend). This was also observed in the nuclear cytoplasmic fraction analysis (Figure 4—figure supplement 2; Figure 4—figure supplement 2 figure, legend). This suggests that KPNA2 might regulate PER2 protein level through another mechanism in the U2 OS model.

Our results in U2 OS cells are different from Umemura et al.’s results in stem cells, probably due to the different cell types and conditions analyzed. We added some discussion in the text to reflect this point (Discussion section).